# Developing a High-Resolution Typical Meteorological Year Dataset for Solar Radiation Evaluation in Australia

Jingpeng Fu<sup>1,2,3,8</sup>, Pingan Ni<sup>1,2,3,8</sup>, Deo Prasad<sup>3</sup>, Guojin Qin<sup>4</sup>, Fuming Lei<sup>1</sup>, Yingjun Yue<sup>1</sup>, Jiaqing Yan<sup>1</sup>, Zengfeng Yan<sup>1,2\*</sup>, Bao-Jie He<sup>5,6,7\*</sup>

<sup>1</sup>School of Architecture, Xi'an University of Architecture and Technology, Xi'an, 710055, China

<sup>2</sup>State Key Laboratory of Green Building, Xi'an, 710055, China

<sup>3</sup>School of Built Environment, University of New South Wales, Kensington 2052, NSW, Australia

<sup>4</sup>School of Ocean and Civil Engineering, Shanghai Jiao Tong University, Shanghai, 200240, China

<sup>5</sup>Centre for Climate-Resilient and Low-Carbon Cities, School of Architecture and Urban Planning, Ministry of Education, Key Laboratory of New Technology for Construction of Cities in Mountain Area, Chongqing University, Chongqing 400045, China

<sup>6</sup>School of Architecture, Design and Planning, The University of Queensland, Brisbane 4072, Australia

CMA Key Open Laboratory of Transforming Climate Resources to Economy, Chongqing, 401147, China

<sup>8</sup>These authors contributed equally

\*Corresponding author:

Zengfeng Yan, Email - yanzengfeng@xauat.edu.cn

Bao-Jie He, Email - baojie.unsw@gmail.com

5

Abstract: High spatiotemporal resolution typical meteorological year (TMY) data are essential for building energy modelling and urban climate studies. However, conventional TMY datasets, limited by sparse ground-based station coverage and infrequent updates, fail to meet the demands of detailed urban-scale simulations. To overcome these limitations, this study uses Australia as a case study and develops a new high-resolution dataset, the TMY derived from the Modern-Era Retrospective analysis for Research and Applications, Version 2 (MERRA-2), hereafter referred to as TMY-MER. A novel weather classification approach was introduced, utilizing a mean relative error index derived from the ratio of daily to monthly maximum solar radiation to identify clear-sky conditions. Uncertainty errors were spatially interpolated using the inverse distance weighting (IDW) method. The results reveal several limitations in the previously generated TMY datasets. TMY-MER demonstrates stable accuracy under clear-sky conditions, with annual average errors below 5%, while under cloudy conditions, influenced by cloud simulation bias, errors can reach up to 50%. Spatially, annual solar irradiance is overestimated by 30% in southeastern coastal urban clusters, while errors in inland regions remain below 10%. Temporally, the peak error during cloudy winter periods reaches 30%, whereas summer clear-sky errors are under 5%. Further analysis using the direct-diffuse separation model indicates a systematic overestimation of diffuse horizontal irradiance (DHI) within 35 6%, and an approximately 20% negative bias in direct normal irradiance (DNI). Validation through building cluster simulations shows that the optimized dataset achieves over 90% consistency with traditional TMY data, with monthly mean errors below 5%. The multidimensional error assessment framework significantly enhances the reliability of reanalysis data for use in complex climate zones, supporting dynamic energy system planning and urban thermal environment modelling. Keywords: atmospheric reanalysis datasets; typical meteorological year; solar radiation; direct-diffuse separation

40

| Nomenclature<br>Abbreviations |                                                       |                      |                                                                                 |
|-------------------------------|-------------------------------------------------------|----------------------|---------------------------------------------------------------------------------|
| AOD                           | Aerosol Optical Depth                                 | LBT                  | Ladybug Tools                                                                   |
| BRL                           | Boland-Ridley-Laurent Model                           | LD                   | Downward longwave radiation                                                     |
| CF                            | Cloud Fraction                                        | MERRA2               | Modern-Era Retrospective Analysis for<br>Research and Applications, Version 2   |
| DBT                           | Dry Bulb Temperature                                  | RH                   | Relative Humidity                                                               |
| DHI                           | Diffuse horizontal irradiance                         | TMY                  | Typical Meteorological Year                                                     |
| DIRINT                        | Direct Insolation Separation Model                    | TMY-MER              | TMY based on MERRA-2 data                                                       |
| DNI                           | Direct normal irradiance                              | TPVs                 | Tibetan Plateau Vortices                                                        |
| GHI                           | Global horizontal irradiance                          | TY                   | Actual year of a TMY                                                            |
| IDW<br><b>Symbols</b>         | Inverse Distance Weighting                            | WS                   | Wind Speed                                                                      |
| $CSD_X^m$                     | Monthly clear sky days in the TMY                     | $MRE_G$              | Mean relative error of global annual data                                       |
| $CSD^m_{MER}$                 | Monthly clear sky days in the MERRA-2                 | $MRE_I$              | Mean relative error between $I_{d,max}$ and $I_{m,max}$                         |
| $CSD^m_{bas}$                 | Baseline number of clear sky days                     | $MRE_L$              | Mean relative error of local monthly data                                       |
| $D_i$                         | Distance                                              | $MRE_O$              | Mean relative error of Observation                                              |
| $D_m$                         | A criterion                                           | p                    | Power of distance                                                               |
| $G_{on}$                      | The corrected solar constant                          | $Q_m$                | Total monthly solar radiation                                                   |
| i                             | Number of parameters                                  | $S_{i,m}$            | Standard error of all years                                                     |
| $I_{d,max}$                   | Daily maximum solar radiation intensity               | $X_{i,m,y}$          | Average value of parameter i of month $m$ in year $y$                           |
| $I_{m,adj}$                   | Adjusted monthly maximum solar radiation intensity    | $\overline{X}_{i,m}$ | Average value of all years                                                      |
| $I_{m,max}$                   | Monthly maximum solar radiation intensity             | $Z_{0}$              | Estimates calculated by spatial interpolation                                   |
| $K_i$                         | Weights of the <i>i</i> -th meteorological parameter  | $z_i$                | Attribute values for (i=1, 2, 3n) samples in the spatial difference calculation |
| $K_t$                         | The clearness index                                   | $\theta_z$           | Zenith angle                                                                    |
| $K_{t}^{'}(t)$                | The modified clearness index at the current time step | $\eta_{i,m,y}$       | Standardized results of the $i$ -th parameter of the $m$ -th month in year $y$  |
| $MRE_A$                       | Mean Relative Error of Adjusted Global Data           | τ                    | Clear-sky MRE <sub>I</sub> threshold                                            |
| $MRE_B$                       | Mean relative error of climate change                 | $\Delta K_{t}^{'}$   | The stability correction index                                                  |

#### 1 Introduction

### 45 1.1 Research background

Solar radiation is the fundamental energy source driving the operation of the Earth's system, and its spatiotemporal distribution profoundly influences the climate system, ecological cycles, and the sustainable development of human society. As the core carrier of renewable energy, solar energy development and utilization not only represent critical pathways for addressing energy crises and climate change but also serves as essential technological support for achieving carbon neutrality targets (Dincer & Aydin, 2023; Olauson et al., 2016; Pfeifroth et al., 2024). Based on solar radiation meteorological datasets, the assessment of solar energy resources can be enabled, and the analysis of energy yields for photovoltaic (PV) and concentrated solar power facilities can be conducted (Bright, 2019; Korany et al., 2016). Moreover, solar radiation is a major factor shaping the urban thermal environment. By forecasting radiative energy, the strategic layout of buildings and green spaces can be optimized to enhance energy efficiency and mitigate the urban heat island effect (Jamei et al., 2020; Khan et 55 al., 2024).

The building sector accounts for nearly 30% of global energy consumption (Falchetta et al., 2024), and accurate, compatible weather data serves as the foundation for building performance simulations (Dong et al., 2021). Simultaneously, reliable solar radiation data are essential for a wide range of scientific activities, including heritage conservation, meteorology, atmospheric sciences, and agricultural applications (Paulescu et al., 2022; Ray et al., 2024; Sarr et al., 2024; R. Wu et al., 2023). Therefore, achieving a more precise assessment of solar radiation data is a fundamental prerequisite for optimizing energy planning and improving the built environment. However, the spatiotemporal distribution of solar radiation is influenced by multiple factors such as geographic location, seasonal variability, atmospheric conditions, and terrain characteristics, leading to inherent complexity and uncertainty in its distribution (Ni et al., 2023). Although mainstream typical meteorological year (TMY) datasets reflect long-term climatic characteristics, they suffer from limitations such as insufficient spatial coverage and delayed updates (Cao et al., 2022). Addressing these evaluation challenges, the construction of meteorological datasets with strong spatiotemporal representativeness has been a critical breakthrough.

## 1.2 Literature review

TMY datasets are constructed by synthesizing long-term meteorological observations to generate a representative artificial year that captures the statistical characteristics of climate variability (Cui et al., 2017). By selecting twelve TMY datasets effectively reflect long-term climate patterns while minimizing the influence of short-term climatic anomalies (Machard et al., 2024; Tambula et al., 2023). These datasets have been widely used to support building energy consumption simulations and solar energy resource assessments by providing reliable climatic inputs (W. Li et al., 2017; Wang, Jia, et al., 2025; Yu et al., 2024). The generation of TMY datasets typically incorporates multiple key meteorological parameters, including dry-bulb temperature, dew-point temperature, and solar radiation (Sun et al., 2020). Various methodologies have 75 been proposed for TMY construction, including the Sandia method (Hall et al., 1978), the Danish method (Lund, 1995), the Festa-Ratto method (Festa & Ratto, 1993), and the China Standard Weather Data (CSWD) method (Song et al., 2007). Prominent global TMY datasets include the U.S. TMY series (Marion & Urban, 1995; Wilcox & Marion, 2008), the International Weather for Energy Calculations (IWEC) series and the CSWD developed by the China Meteorological Administration (Song et al., 2007). Nevertheless, existing TMY datasets are often outdated and suffer from insufficient temporal resolution, limiting their ability to meet the growing demand for high-accuracy, real-time meteorological information required by modern building performance simulations and dynamic solar energy assessments.

The effectiveness of the Sandia method for TMY generation has been validated in solar energy assessments in Turkish building energy consumption simulations across multiple climatic zones (H. Li et al., 2023; Pusat et al., 2015). Concurrently, national datasets have been successively released in countries such as Argentina (Bre & Fachinotti, 2016), Australia (Ren et

al., 2021), and Canada (Berardi & Jafarpur, 2020). Despite their widespread adoption, conventional TMY methods continue to face several important limitations. In particular, most existing TMY datasets rely heavily on ground-based meteorological observations. However, the sparse distribution of observations (Cao et al., 2020; Hao et al., 2020), combined with the widespread lack of solar radiation sensors (Bre et al., 2021; Cao et al., 2022), significantly constrains their spatial coverage and reduces their ability to represent regional climatic variability.

To overcome these spatial and temporal limitations, reanalysis datasets have become an increasingly important resource in recent years. Leveraging advances in data assimilation techniques, these gridded datasets integrate multiple sources of measurements with numerical model outputs, thereby improving spatial resolution and temporal continuity (Abatzoglou, 2013; Tang et al., 2019). Against these, the MERRA-2 reanalysis dataset developed by NASA's Global Modelling and Assimilation Office (NASA GES DISC, 2025) has emerged as a benchmark in the field due to its technological breakthroughs. Its hourly temporal resolution and improved assimilation algorithms, which incorporate aerosol data from AVHRR, MODIS and MISR satellites, and AERONET ground stations, substantially enhance the simulation accuracy of climate system interactions (Randles et al., 2017). MERRA-2 has been extensively applied across a wide range of disciplines, including climate dynamics research (e.g., diagnoses of tropical cyclone dissipation mechanisms and interdecadal evolution of Tibetan Plateau vortices) (Lin et al., 2024; Xie et al., 2024); land surface process modelling (e.g., permafrost model optimization and ocean surface net radiation calculations) (Liang et al., 2022; Qin et al., 2020), and the characterization of extreme weather events (Chen et al., 2024; Horan et al., 2024; Pulliainen et al., 2020). These applications demonstrate the significant multidisciplinary research value of MERRA-2. This paper primarily analyzes the solar radiation data from the dataset.

Nonetheless, while MERRA-2 has filled many observational gaps, its radiation data are not without flaws. Studies have shown that these biases primarily originate from deficiencies in simulating aerosol–cloud interactions. On the one hand, assimilation errors in aerosol optical depth (AOD) (Chakraborty & Lee, 2021) and the underestimation of low cloud cover (Camberlin et al., 2023) contribute to the overestimation of solar radiation. On the other hand, inadequate parameterization of cloud microphysical processes exacerbates radiation estimation errors over complex terrains (e.g., the Tibetan Plateau) and monsoon regions (Qi et al., 2024). Although researchers have identified deviations in radiation parameters through point-based validations (Aliana et al., 2022; Feng et al., 2023), most existing studies are constrained by short temporal records or small sample sizes (Du et al., 2022; Isaza et al., 2024), limiting their ability to capture the spatiotemporal heterogeneity of errors.

Efforts to validate MERRA-2 radiation data have achieved some regional progress, but several limitations remain unresolved. In China, researchers have consistently reported an overestimation of global horizontal irradiance (GHI), largely due to underrepresentation of cloud cover and AOD in the model (Cao et al., 2022; Du et al., 2022). Yet, these findings are often based on single-year analyses or long-term assessments that do not differentiate between weather types. In contrast, studies in Australia have highlighted an overestimation of AOD and an underestimation of solar radiation (Isaza et al., 2024), though sparse observational networks limit broader spatial conclusions. These discrepancies underscore the need for more comprehensive validation studies that span diverse regions, extended time periods, and varying meteorological conditions in order to improve more accuracy and applicability of MERRA-2 radiation data.

# 1.3 Research innovations

This study innovatively integrates reanalysis data with TMY generation methods, using MERRA-2 as the primary data source to develop TMY-MER, a nationwide TMY dataset for Australia. Through a spatiotemporal error decomposition model, the multidimensional distribution patterns of solar radiation errors in TMY-MER are systematically characterized.

The major innovations and contributions of this study are:

- (1) Development of a TMY dataset with high spatial coverage. Based on the most recent 15 years of MERRA-2 data, TMY-MER overcomes the limitations of traditional TMY datasets that rely on sparse ground-based observations, providing timely and high-resolution support for dynamic energy planning and urban thermal environment modelling.
- (2) Multidimensional solar radiation error analysis. A novel weather-type classification method based on clear-sky identification is proposed to separately analyse solar radiation errors under clear and overcast conditions, revealing the spatiotemporal distribution patterns of errors. Coupled with inverse distance weighting (IDW) interpolation, the spatial generalization of error parameters is achieved, enabling the optimization of solar radiation data across all TMY-MER sites.

Figure 1. Research framework

#### 2 Methods and materials

Figure 1 illustrates the core framework of this study, which aims to construct a high-accuracy dataset, named TMY35 MER, encompassing three major modules: data generation, error assessment, and application validation.

**Step 1:** Based on MERRA-2 reanalysis data for hourly meteorological parameters for 2009 to 2023, from 353 sites across Australia were extracted. Using a TMY generation method that combines parameter normalization and weighted calculation, twelve TMMs were selected from the historical multiyear dataset. A cubic spline interpolation model was applied to smooth transitions at monthly boundaries, thereby enhancing data continuity and constructing a TMY-MER dataset covering all county-level regions in Australia.

Step 2: To address the systemic errors in MERRA-2 solar radiation data, a classification-based assessment method incorporating clear-sky identification was proposed. Clear and cloudy days were distinguished based on the relative error (MREI) between the daily maximum solar radiation intensity  $(I_{d,max})$  and the monthly maximum solar radiation intensity  $(I_{m,max})$ . Monthly adjustments were applied to the radiation intensity on clear days, while the radiation data for cloudy days were adjusted based on the monthly total radiation. To ensure spatial coverage accuracy, IDW interpolation was employed to generalize the error parameters, enabling error correction across 353 TMY-MER sites in Australia.

Step 3: The accuracy of the adjusted data was validated by simulating building surface solar radiation on the Ladybug Tools (LBT) platform, comparing the results between the original TMY dataset and the adjusted TMY-MER dataset. The DIRINT model was applied to separate the GHI into direct normal irradiance (DNI) and diffuse horizontal irradiance (DHI). Additionally, a multi-condition sky model database was further developed to enable efficient evaluation of solar energy distribution under various complex environmental scenarios.

## 2.1 Generation of TMY-MER

#### 2.1.1 Data sources

MERRA2 is a new-generation reanalysis dataset, with spatiotemporal coverage extending from 1980 to the present. It offers a spatial resolution of  $0.5^{\circ} \times 0.625^{\circ}$  and a temporal resolution as fine as one hour (Gelaro et al., 2017). By assimilating bias-corrected satellite aerosol observations and clear-sky solar radiation data and using observed precipitation to drive land surface process simulations, MERRA-2 significantly enhances the ability to accurately reproduce atmospheric parameters. Its core algorithms integrate the Chou-Suarez shortwave radiation scheme and the latest satellite data assimilation techniques from the GEOS-5 system (Randles et al., 2017). The total aerosol optical depth (AOD) of the dataset has been independently validated and is shown to be reliable (Buchard et al., 2017). In this study, 15 consecutive years of data from 2009 to 2023 were selected as the foundational dataset for constructing the TMY database. The key meteorological variables extracted from MERRA-2 for developing a high-resolution, multi-dimensional solar radiation dataset include: average dry-bulb temperature ( $T_{avg}$ ), maximum dry-bulb temperature ( $T_{max}$ ), minimum dry-bulb temperature ( $T_{min}$ ), mean total column water vapor ( $Q_{avg}$ ), maximum total column water vapor ( $Q_{max}$ ), minimum total column water vapor ( $Q_{min}$ ), average relative humidity ( $RH_{avg}$ ), average wind speed ( $V_{avg}$ ), and hourly GHI.

# 2.1.2 Method for generating the TMY-MER

This study focuses on the Australian region, using county-level boundaries as the basis for spatial segmentation. Hourly meteorological data from the past 15 years (2009–2023) were extracted from the MERRA-2 reanalysis database for approximately 353 weather stations across these counties, including solar radiation measurements. Based on this dataset, the data were processed using well-established methodologies for generating TMY datasets. A recent approach to TMY construction, originally developed for ERA5 reanalysis data, was adapted in this study to select 12 meteorological months

from historical multi-year records, forming a complete annual weather sequence for each station (Y. Wu et al., 2023). The detailed selection procedure for these 12 typical months is outlined below.

- (1) Parameter selection and monthly mean calculation: Seven key parameters were extracted from the AMY dataset spanning 2009 to 2023. For each parameter (i), the monthly value  $(X_{i,m,y})$  was calculated for each month (m) and year (y).
  - (2) Normalization of monthly values: Based on the 15-year dataset, the mean  $(\overline{X}_{i,m})$  and standard error  $(S_{i,m})$  were computed for each parameter and month. Monthly values were then standardized using Equation (Eq.) (1) to obtain the normalized values  $((\eta_{i,m,y}))$ .
- (3) Selection of typical meteorological months as candidates: A month was considered a typical candidate month if the normalized values (η<sub>i,m,y</sub>) for all seven parameters were less than 1. Multiple candidate months could be identified under this criterion.
  - (4) Weighted score calculation and selection of typical meteorological months: Weights were assigned to the seven parameters based on their relative importance (Table 1). For each candidate month, a weighted score  $(D_m)$  was calculated using Eq. (2). The month with the lowest score was subsequently selected as the typical meteorological month.

**Table 1.** Factors and their weights for selection

| Meteorological Parameter | Selection parameter                          | Weight |  |
|--------------------------|----------------------------------------------|--------|--|
|                          | $T_{avg}$ (Average dry-bulb temperature)     | 2/20   |  |
| Dry-bulb temperature     | $T_{max}$ (Maximum dry-bulb temperature)     | 1/20   |  |
|                          | $T_{min}$ (Minimum dry-bulb temperature)     | 1/20   |  |
|                          | $Q_{avg}$ (Average total quantity of vapors) | 2/20   |  |
| Total quantity of vapors | $Q_{max}$ (Maximum total quantity of vapors) | 1/20   |  |
|                          | $Q_{min}$ (Minimum total quantity of vapors) | 1/20   |  |
| Relative humidity        | $RH_{avg}$ (Average relative humidity)       | 1/20   |  |
| Wind speed               | $V_{avg}$ (Average wind speed)               | 1/20   |  |
| Solar radiation          | GHI( Global horizontal irradiance)           | 10/20  |  |

(5) Smoothing at monthly boundaries: To address discontinuities at the transitions between the 12 selected months, a cubic spline interpolation model (McKinley & Levine, n.d.) was applied to the 24-hour interface data. The original 24 interface values were replaced by a linearly distributed set of smoothed values.

The resulting TMY-MER station distribution demonstrates strong spatial representativeness. Each county-level region in Australia is covered by at least one meteorological station. Moreover, areas with higher urban density feature finer subdivisions and denser TMY-MER station distributions, ensuring that the extracted datasets more accurately capture the full meteorological characteristics of each region.

$$\eta_{i,m,y} = \frac{X_{i,m,y} - \overline{X}_{i,m}}{S_{i,m}} \tag{1}$$

where *i* is the number of parameters, *m* is the number of months, and *y* is the number of years,  $X_{i,m,y}$  is the average value of parameter *i* in month *m* of year *y*,  $\overline{X}_{i,m}$  is the average value of all years, and  $S_{i,m}$  is the standard error of all years.

$$D_{m} = \sum_{i=0}^{n} K_{i} \left| \eta_{i,m,y} \right| \tag{2}$$

where  $D_m$  is a criterion,  $K_i$  is the weight of the *i-th* meteorological parameter, and  $\eta_{i,m,y}$  are the standardized results of the *i-th* parameter of the *m-th* month in year y, as shown in Eq. (1).

#### 2.2 Multi-dimensional error assessment

Although ERA-5 and MERRA-2 data are widely used in many studies due to their advantages, such as high spatial resolution, finer temporal granularity, and long time series, existing research has shown that reanalysis data may contain errors that significantly exceed those caused by climate change. This study further evaluates the multidimensional errors in GHI between TMY-MER and traditional TMY datasets, defining potential climate change and observational errors based on different weather types. To address the multidimensional errors in MERRA-2 data, the clear-sky error, which is relatively small and well-defined, is set as the baseline error ( $MRE_B$ ). After identifying the baseline errors caused by climate change, they are further used to assess the highly uncertain overcast day errors. The error assessment process mainly includes the following five steps:

Step 1: Weather type classification was based on clear-sky identification. Widely available, low-resolution meteorological data were selected as baseline data, with TMY serving as an example in this study. First, the relative error  $(MRE_I)$  between daily maximum solar radiation intensity  $(I_{d,max})$  and monthly maximum solar radiation intensity  $(I_{m,max})$  was calculated. Days with an error below a predefined threshold  $(\tau)$  were classified as clear-sky days, and the number of clear-sky days for both datasets was denoted as  $CSD_X^m$  and  $CSD_{MER}^m$ . These values were ranked according to  $MRE_I$  size, and the corresponding  $MRE_I$  and daily sequences were recorded.

Step 2: Monthly clear day error quantification. The smaller value between  $CSD_X^m$  and  $CSD_{MER}^m$  for each month was selected as the reference number of clear days, denoted as  $CSD_{bas}^m$ . The relative error between the monthly average solar radiation values of the two datasets was evaluated. Radiation discrepancies exceeding the climate change baseline error  $(MRE_B)$  were defined as clear-sky observation errors  $(MRE_O)$ . Based on  $MRE_O$ , the selected clear day data were either reduced or increased to ensure that the monthly maximum radiation on clear days in TMY-MER remained within the bounds defined by  $MRE_B$ . The recalibrated monthly maximum solar radiation intensity  $(I_{m,adj})$  was then recalculated for subsequent anomaly detection.

Step 3: Annual and monthly overcast day error calculation. Due to the high uncertainty in solar radiation intensity  $(I_{d,max})$  under overcast conditions, the total monthly solar radiation  $(Q_m)$  was adopted as a more robust metric for error assessment. Local monthly relative errors  $(MRE_L)$  and overall annual relative errors  $(MRE_G)$  were evaluated beyond the baseline error  $(MRE_B)$ .  $MRE_L$  was used to scale solar radiation for overcast days, ensuring better consistency in monthly totals, while  $MRE_G$  was retained to preserve the intermonth solar radiation differences between TMY-MER and TMY datasets, reflecting the effects of climate change.

Step 4: Error anomaly detection and adjustment. To avoid the situation where  $CSD_X^m$  exceeds  $CSD_{MER}^m$  but requires a substantial upward adjustment of TMY-MER for cloudy days, which could result in excessively high sunny day solar radiation intensities outside the  $CSD_{bas}^m$  threshold, anomaly detection was implemented. For each month, the original  $I_{d,max}$  were re-evaluated against the  $I_{m,adj}$ . If any  $I_{d,max}$  exceeded  $I_{m,adj}$ , the adjusted hourly averages from the revised  $CSD_{bas}^m$  dataset were substituted. Following the anomaly detection and adjustment process, the  $MRE_T$  was reassessed.

Step 5: Spatial interpolation of error parameters. Error parameters derived from low-resolution station comparisons were interpolated to enhance the spatial accuracy of the TMY-MER dataset. IDW interpolation (Ding et al., 2025; Wang, Yao, et al., 2025; Yanto et al., 2017), as defined in Eqs. (3) and (4), was applied to calculate the weighted average of error parameters from N neighbouring reference stations. The spatial distributions of  $CSD_{bas}^{m}$ ,  $MRE_{B}$ ,  $MRE_{L}$ ,  $MRE_{G}$  and  $MRE_{T}$  were then computed for all stations.

$$z_o = \sum_{i=1}^n \frac{1}{(D_i)^p} z_i \left[ \sum_{i=1}^n \frac{1}{(D_i)^p} \right]^{-1}$$
 (3)

$$D_i = \sqrt{(x_o - x_i)^2 + (y_o - y_i)^2}$$
 (4)

where  $z_o$  represents the estimated value,  $z_i$  denotes the observed attribute at sample point i,  $D_i$  is the distance between the target location and the sample point, and p is the power parameter controlling the smoothing effect, typically set to 2 to minimize the mean absolute error.

# 2.3 Data validation and scalability

To verify the accuracy and reliability of the adjusted solar radiation data derived from the TMY-MER dataset, two building clusters characterized by complex shading relationships or unique layouts were selected as research subjects. These clusters encompass diverse building types, height variations, and spatial configurations, effectively simulating the varied impacts on solar radiation observed in real urban environments. Using the LBT platform (Ni et al., 2023; Yue et al., 2024), surface solar radiation simulations were conducted for the selected clusters using both TMY meteorological data and the corresponding TMY-MER data. During the simulations, the distribution characteristics of solar radiation on building surfaces were meticulously recorded and compared. In parallel, sky dome models were generated for each meteorological dataset, and a comparative analysis of their luminance distributions was conducted. This analysis systematically examined luminance variations across different azimuths and elevation angles, as well as trends in overall luminance values. To enhance computational efficiency and meet the demands of complex simulations, a multi-dimensional sky model database was developed, introducing an innovative data storage structure to keep vectorized parameters and solar radiation matrices. This module can parse solar radiation data from weather files, invoke the Radiance engine, and process complex simulation scenarios involving multiple temporal dimensions and radiation types in batches, thereby meeting the demands of advanced simulations.

To address the limitation of MERRA-2, which only includes GHI data, this study utilized the DIRINT model (Ineichen et al., 1992) from the pylib library (https://github.com/pylib/pylib-python) as a method for splitting total radiation into direct, normal, and horizontally diffuse components. By introducing the  $\Delta K_t$  stability index, the model can adaptively adjust the estimated DNI based on the dynamic characteristics of the GHI time series, thereby enhancing its adaptability under varying weather conditions, as shown in Eqs. (5)-(7). The DIRINT model is suitable for a range of environmental conditions, including different pressure and temperature conditions, making it widely applicable globally (Engerer, 2015; Lave et al., 2015).

$$K_{t} = \frac{GHI}{G_{on} \cdot cos(\theta_{z})}$$

$$DNI = \frac{GHI - DHI}{cos(\theta_{z})}$$
(6)

$$DNI = \frac{GHI - DHI}{\cos(\theta_z)} \tag{6}$$

$$\Delta K_t' = K_t'(t) - K_t'(t-1) \tag{7}$$

where  $K_t$  is the clearness index, GHI is the global horizontal irradiance,  $G_{on}$  is the corrected solar constant, and  $\theta z$  is the solar zenith angle. DNI is the direct normal irradiance, DHI is the diffuse horizontal irradiance.  $\Delta K_t$  is the stability correction index,  $K_t(t)$  is the modified clearness index at the current time step, and  $K_t(t-1)$  is the value at the previous time step.

# 2.4 Research area

Australia was selected as a representative study region due to its unique resource conditions and the requirements of the research design. One of the main reasons is the high photovoltaic development potential in low-density urban areas. The dispersed building layout in these areas significantly reduces shading effects, thereby increasing the effective operating hours and unit-area efficiency of photovoltaic systems. Studies have shown that the photovoltaic potential in low-density areas can reach 1 to 6.6 times total energy consumption, which is significantly higher than that in medium- to high-density urban areas (Geng et al., 2024).

Another important consideration is the need to verify climate and geographic diversity. Australia spans from tropical to cool temperate climates and features a variety of terrains, including coastal plains, mountains, and deserts (Hughes, 2011).

Its climate and geographic characteristics, such as sizeable convective clouds in the tropical monsoon region, stratocumulus clouds in the temperate oceanic climate, and clear-sky conditions in desert regions) provide an effective means to validate the applicability of reanalysis data in different environments, enhancing the universality of the error assessment methods.

A further justification for the selection of Australia is the urgency of energy transition and the availability of supporting data. As one of the countries with the richest solar resources in the world (with an average annual solar radiation of 2000–2500 kWh/m²), the Australian government plans to achieve a renewable energy share of 82% by 2030 (Yoo, 2023). High-precision solar radiation data is crucial for photovoltaic system planning and grid scheduling, and the scarcity of traditional ground-based observation data further underscores the value of constructing the TMY-MER dataset in this study. To support the subsequent analysis with a spatial context, geographic illustrations are employed in the following sections. All maps of Australia used in this study are sourced from OpenStreetMap (Herfort et al., 2023).

# 285 3 Results

# 3.1 Data sample documentation

Table 2. Sample of TMY-MER data records for Bega Valley, AUS

| MONTH | DAY | HOUR | DBT    | RH     | WS      | WD    | GHI       | AMY  |
|-------|-----|------|--------|--------|---------|-------|-----------|------|
|       |     |      | (°C)   | (%)    | (m/s)   | (°)   | $(W/m^2)$ | (y)  |
| 1     | 1   | 0    | 16.205 | 35.388 | 249.906 | 2.983 | 0         | 2015 |
| 1     | 1   | 1    | 16.549 | 36.583 | 251.552 | 2.926 | 0         | 2015 |
| 1     | 1   | 2    | 16.666 | 37.435 | 252.302 | 2.764 | 0         | 2015 |
| 1     | 1   | 3    | 16.730 | 37.865 | 254.672 | 2.548 | 2.330     | 2015 |
| 1     | 1   | 4    | 16.758 | 37.944 | 261.009 | 2.487 | 104.781   | 2015 |
| 1     | 1   | 5    | 17.242 | 37.699 | 268.873 | 2.250 | 311.625   | 2015 |
| 1     | 1   | 6    | 18.626 | 36.774 | 273.367 | 2.053 | 529.250   | 2015 |
| 1     | 1   | 7    | 20.608 | 34.667 | 277.232 | 1.832 | 726.750   | 2015 |
| 1     | 1   | 8    | 22.397 | 32.984 | 279.595 | 1.114 | 888.750   | 2015 |
| 1     | 1   | 9    | 23.585 | 31.771 | 267.137 | 0.804 | 1000.000  | 2015 |
| 1     | 1   | 10   | 24.584 | 30.769 | 237.193 | 0.795 | 1054.500  | 2015 |
| 1     | 1   | 11   | 24.931 | 31.477 | 206.241 | 0.945 | 1048.500  | 2015 |
| 1     | 1   | 12   | 24.814 | 32.454 | 183.234 | 1.292 | 976.250   | 2015 |
| 1     | 1   | 13   | 24.354 | 33.058 | 166.891 | 1.829 | 841.250   | 2015 |
| 1     | 1   | 14   | 23.572 | 33.659 | 152.768 | 2.642 | 647.500   | 2015 |
| 1     | 1   | 15   | 22.462 | 34.374 | 142.915 | 3.087 | 419.500   | 2015 |
| 1     | 1   | 16   | 21.184 | 34.677 | 135.434 | 3.116 | 200.438   | 2015 |
| 1     | 1   | 17   | 19.943 | 34.475 | 131.307 | 2.857 | 57.766    | 2015 |
| 1     | 1   | 18   | 18.906 | 34.060 | 125.832 | 2.496 | 1.343     | 2015 |
| 1     | 1   | 19   | 18.327 | 34.016 | 116.899 | 2.061 | 0         | 2015 |
| 1     | 1   | 20   | 18.033 | 33.948 | 113.845 | 1.672 | 0         | 2015 |
| 1     | 1   | 21   | 17.793 | 33.928 | 108.520 | 1.245 | 0         | 2015 |
| 1     | 1   | 22   | 17.511 | 33.873 | 99.784  | 0.979 | 0         | 2015 |
| 1     | 1   | 23   | 17.321 | 33.826 | 90.687  | 0.775 | 0         | 2015 |

A total of 353 TMY-MER data sites were generated for the Australian region. The grid was selected based on the open-source World Cities Database, which covers major cities in Australia. The geographic distribution of the TMY-MER dataset spans 6 states and 2 major territories, covering a total of 353 local government areas in Australia. For each grid, the TMY includes data for 8,760 hours. The complete meteorological variables include month, day, hour, dry bulb temperature (°C), relative humidity (%), wind speed (m/s), wind direction (°), global horizontal irradiance (W/m²), and the actual year corresponding to the typical meteorological month (y). The complete data records for TMY-MER are listed in Table 2. Additionally, the weather data was converted to the EPW format for building performance simulation.

#### 3.2 Evaluation of relative errors in total solar radiation

Figure 2. Error distribution of TMY-MER solar radiation data in Australia.

This study extracted the TMY solar radiation data from 122 TMY sites in Australia. The corresponding nearest neighbor TMY-MER data sites were selected for each TMY site. The TMY-MER solar radiation data generated were compared with the corresponding TMY data. The annual total errors and the error distribution for each month are shown in Fig. 2. Regarding the total solar radiation over the entire year, although the MERRA-2 dataset exhibits significant geographical variability in its error distribution, the error magnitude remains within 15%. Monthly time series analysis reveals clear seasonal variations in error characteristics. During the Australian summer (December to February), radiation errors exhibit notable spatial heterogeneity. The TMY-MER dataset consistently shows a positive bias along the eastern, southern, and southwestern coasts, while the northern and northwestern coastal regions display negative biases, with the maximum reaching 30%. In the inland areas of Australia, the radiation error magnitude is relatively low and predominantly

positive, with the vast majority of sites showing positive bias not exceeding 10%. In the transitional seasons of spring and autumn, March and November (months approaching summer) in the northern and northwestern coastal areas still show negative bias, while other months exhibit varying degrees of positive bias.

The spatial distribution pattern is as follows: the error magnitude is largest in the eastern and southern coastal areas, with autumn showing higher error magnitudes than spring; the inland areas exhibit smaller error magnitudes, not exceeding 10%. During winter (June to August), the TMY-MER dataset shows positive bias in all monitored regions, with the largest error magnitude observed in the northwest coastal region (with an extreme value of 30%), followed by the southern coastal regions. The central region has a significantly higher error magnitude compared to other seasons, reaching more than 20%. It is noteworthy that the positive bias in the southwestern coastal region during winter is notably lower than the error levels in this region during the summer. A comparison of the spatial and temporal scales reveals that more urbanized areas, such as the southeastern coastal region of Australia and the southwestern part of Western Australia (WA), exhibit more pronounced positive bias characteristics.

# 3.3 Weather type identification

**Figure 3.** Hourly solar radiation on clear days in Central Highlands, Tasmania (kWh/m²) Note: Each curve represents the hourly distribution of solar radiation on a particular clear day.

A clear-sky detection algorithm was applied to classify solar radiation parameters from the TMY and TMY-MER meteorological datasets into clear and overcast weather types. To ensure more accurate identification of monthly clear days at each site, a specialized classification model was developed. The study analyzed 122 geographically diverse locations across Australia, using TMY as the primary dataset and TMY-MER as the reference. For each site, monthly statistics were compiled for the number of clear and overcast days, along with corresponding solar radiation data. Hourly solar radiation distribution curves were constructed for each day of the month, allowing for comparative analysis of radiation patterns using

clustering methods. Site P38, located in the Central Highlands of Tasmania (TAS), was selected as a representative example (Fig. 3 and 4).

From a temporal perspective, the monthly radiation curves exhibit clear seasonal variation. In summer, particularly in December and January, the daily radiation distribution range expands significantly, with peak intensities approaching 1000 kWh/m². In contrast, during winter in June and July, daily peak values typically fall below 600 kWh/m², indicating strong alignment between solar radiation variability and seasonal transitions. Weather-type analysis further reveals that, within the same month, clear days consistently exhibit higher solar radiation intensities than overcast days. The hourly radiation curves on clear days are more concentrated and stable, whereas those on overcast days are highly dispersed and variable. For instance, in December, the maximum daily radiation difference between TMY and TMY-MER under overcast conditions exceeds 500 kWh/m², indicating substantial intraday fluctuations. Moreover, although these curves generally follow a diurnal cycle, overcast conditions introduce pronounced nonlinearity due to dynamic cloud interactions affecting solar transmittance.

**Figure 4.** Hourly solar radiation on overcast days in Central Highlands, Tasmania (kWh/m²) Note: Each curve represents the hourly distribution of solar radiation for a particular overcast day.

Annual comparative results show that TMY-MER performs reliably under clear-sky conditions, with radiation estimates exhibiting a positive bias of less than 10% relative to TMY. However, under overcast conditions, particularly during the summer months (December to February), the bias increases significantly, reaching up to 30-40%. Despite overall consistency between the datasets in the proportion of weather types, notable classification discrepancies emerge during certain periods. For example, at site P38 in August and September (late winter and early spring), significant differences are observed between the datasets. Based on these findings, subsequent site-level error analyses will adopt the monthly average solar radiation under each weather type as the baseline parameter.

# 3.4 Error analysis by weather type based on clustering

#### 3.4.1 Monthly errors under different weather conditions

Based on the established evaluation methodology, monthly solar radiation errors were calculated at each site using average values under clear and overcast conditions as reference baselines. From the complete dataset, 24 representative meteorological stations were selected to analyse the distribution characteristics of monthly average solar radiation in the TMY-MER dataset under both weather scenarios as illustrated in Figs. 5 and 6. Under clear-sky conditions, TMY-MER consistently exhibits a slight positive bias across most sites, with error characteristics closely resembling those of site P38. Sites P36 and P37 show notably higher levels of overestimation, with annual cumulative errors exceeding 5%. These systemic deviations are mainly attributed to the stable optical properties of the atmosphere during clear weather, where minimal cloud cover and low aerosol concentrations lead to seasonal radiation patterns that govern the bias. Consequently, monthly errors at individual sites remain temporally consistent and spatially uniform, indicating robust model performance under clear-sky conditions.

Figure 5. Monthly solar radiation errors across different regions under clear-sky conditions at various sites. (kWh/m²) Source

In contrast, solar radiation errors under overcast conditions are more complex and variable. Greater cloud cover uncertainty, increased aerosol loading, and diverse cloud configurations exert significant influence on radiative transfer, resulting in larger fluctuations in the discrepancies between TMY-MER and TMY datasets. Substantial overestimations exceeding 40% are observed during specific months at several stations, including P13 (March, August, December), P21

(November), P33 (November), and P36 (January). These findings highlight the sensitivity of solar radiation estimates to atmospheric variability under cloudy conditions and suggest that TMY-MER's performance is more susceptible to local meteorological disturbances in such scenarios.

**Figure 6.** Monthly analysis of solar radiation errors across different regions under overcast conditions at various sites (kWh/m²).

Spatial analysis reveals pronounced regional differences in the distribution of solar radiation estimation errors. In the western coastal regions—such as stations P27, P36, and P37—TMY-MER consistently overestimates solar radiation, particularly during the wet season from November to March. These overestimations are likely linked to seasonal cloud variability and aerosol emissions in these areas. In the central inland regions—including but not limited to P13, P16, and P17—discrepancies between TMY-MER and reference datasets remain relatively small under clear-sky conditions, due to limited anthropogenic activity and low urban density, which reduce the influence of cloud cover and aerosols on reanalysis products such as MERRA-2. However, during cloudy winter months, notable overestimations are observed, and seasonal variations in bias are evident. For example, site P13 shows substantial overestimation in December and March under cloudy conditions, while P17 exhibits slight underestimation during spring and summer. At northern sites—namely P11, P18, and P28—the TMY-MER tends to overestimate solar radiation during summer, while in other months, most sites exhibit underestimation. For instance, P11 shows underestimation in autumn; P18 and P23 exhibit underestimation in both summer and autumn; P19 underestimates during spring and summer; and P28 underestimates from October to December. In southern regions, locations such as P4, P14, and P27 show varying degrees of overestimation of solar radiation during summer and

adjacent months under overcast and rainy conditions, whereas the errors in solar radiation during winter are relatively minor.

These spatial patterns highlight the heterogeneity of TMY-MER performance across climatic zones and emphasize the influence of local atmospheric conditions on radiation modelling accuracy.

# 3.4.2 Spatiotemporal distribution of high-resolution solar radiation errors

**Figure 7.** Spatiotemporal distribution of high-resolution solar radiation errors after IDW interpolation; (a) Annual solar radiation error on clear days; (b) Monthly solar radiation error on clear days; (c) Annual solar radiation error on cloudy and rainy days; (d) Monthly solar radiation error on cloudy and rainy days; (e) Annual solar radiation error error assessment at station P37.

By comparing solar radiation data from 122 TMY stations and their corresponding TMY-MER counterparts across Australia, solar radiation biases under clear-sky and overcast conditions were derived using a cluster-based weather-type classification. Using the IDW method, a weighted mean error was computed for each TMY-MER site based on its nearest bias-adjusted reference stations. This enabled the derivation of the spatial distribution of solar radiation errors across all 353 TMY-MER stations under both weather scenarios, as illustrated in Fig. 7a and 7c. Weather-type based annual radiation

analysis revealed that, under clear-sky conditions, the relative bias between the TMY-MER and TMY datasets ranged from 0% to 6%, with TMY-MER consistently exhibiting a positive bias. Spatially, smaller errors were found in the northwestern Cape York Peninsula, the southwestern region, and the eastern coastal belt between 30°S and 40°S. In contrast, larger deviations appeared in central Cape York, southern South Australia (SA), southern Victoria (VIC), TAS, and arid interior regions. Under overcast conditions, the bias range expanded to between -40% and 40%, with negative biases dominating the tropical north, TAS, and most inland areas, while positive biases were observed along the southwestern and eastern coasts, as well as in southeastern VIC and SA.

Temporal analysis, as illustrated in Fig. 7b and 7d, further reveals clear seasonal patterns in the spatial distribution of bias. Under clear-sky conditions from April to August, TMY-MER exhibited mild positive deviations, generally within 5%, across southeastern and northern coastal regions, while southwestern coastal areas showed negative biases reaching up to 400 10%. Between October and December, radiation biases were mostly negative and of smaller magnitude, with larger deviations concentrated in coastal zones and smaller ones inland. In February, biases approached zero across most regions, except for a localized positive deviation of approximately 5% over Cape York and the northern part of the NT. Under overcast conditions, seasonal bias variability became more pronounced. During summer, areas north of the Tropic of Capricorn experienced substantial negative biases ranging from -30% to -40%, whereas southern regions generally exhibited positive biases, with coastal zones showing more prominent errors compared to inland areas. In winter, the southwestern coastal region of Australia, including TAS, showed negative biases between -20% and -30%, while most other regions displayed positive biases, again with greater deviations along the coast. During transitional seasons, widespread positive biases ranging from 20% to 40% were observed throughout most coastal regions, excluding northern Australia, whereas inland areas generally exhibited smaller or slightly negative deviations. Validation at a representative station, site P37 (Fig. 7e), confirmed that the adjusted TMY-MER dataset effectively corrected the systematic biases observed in the original dataset across all 8,760 hourly time steps throughout the year.

# 3.5 Regional mean biases

Figure 8a presents the mean annual solar radiation bias across Australian administrative regions, comparing two meteorological datasets: TMY-MER and TMY. Figure 8b further evaluates the total radiation errors under clear-sky and overcast conditions, based on a sky condition classification approach. Figure 8c and 8d display the monthly distributions of regional mean solar radiation bias under these two distinct weather types, respectively. Under clear-sky conditions, the TMY-MER dataset generally overestimates solar radiation, with a distinct spatial pattern characterized by larger biases along the coastal regions and smaller biases in inland areas. Temporally, while monthly error trends remain relatively consistent, pronounced seasonal variations are evident across different regions. For instance, in inland zones such as the 'Outback' and Anangu Pitjantjatjara Yankunytjatjara Lands in SA, and the Petermann-Simpson region in the NT, the positive bias during winter is markedly reduced compared to other seasons. Conversely, in certain eastern coastal regions of WA, including Irwin, the model exhibits a more significant overestimation during winter; however, the maximum underestimation remains below 15%. Notably, a limited number of regions exhibit negative biases during specific months. For example, in February, the Cape York area in Queensland (QLD) shows a slight underestimation, as does the western subregion of the Victoria Desert in SA and the Corangamite, Ararat Surrounds, and St Arnaud areas in VIC during August. However, in all these cases, the maximum underestimation remains below 15%.

The monthly bias patterns under overcast conditions reveal more complex spatial and seasonal dynamics. During the summer months (e.g., December and February), substantial underestimations, up to 60%, are observed in the northern coastal areas, such as Kununurra and Halls Creek in WA and the Victoria River region in the NT. In contrast, central regions display relatively minor deviations. In winter and transitional months (April to October), moderate positive biases occur in eastern coastal QLD, eastern TAS, and southwestern WA, while most other regions are dominated by significant negative biases.

Specifically, underestimations in northern coastal and adjacent inland regions during the transitional months range from 30% to 40%, exceeding those observed in winter alone. In southeastern coastal areas, winter underestimation reaches its peak, with values up to 45%, surpassing those in the transitional season as well.

**Figure 8.** Distribution of average errors across district and county boundaries; (a) Annual solar radiation error; (b) Annual solar radiation error for clear days and cloudy/rainy days following weather type clustering; (c) Monthly solar radiation error on clear days; (d) Monthly solar radiation error on cloudy and rainy days.

# 3.6 Direct-diffuse decomposition and error analysis

After constructing and precisely adjusting the TMY-MER dataset, the DIRINT model was applied to decompose GHI into DNI and DHI. Figure 9a illustrates the decomposition at site P1814, which is located in the Bega Valley in New South Wales (NSW), and produced a complete 8,760-hour time series of DNI and DHI derived from the TMY-MER solar radiation data. Both DNI and DHI exhibit pronounced seasonal variability, with summer peaks reaching approximately 900 kWh·m<sup>-2</sup>

and 500 kWh·m<sup>-2</sup>, respectively, while winter maxima decrease to around 700 kWh·m<sup>-2</sup> and 400 kWh·m<sup>-2</sup>. Figure 9b presents the hourly decomposition results for January and July using the DIRINT model, alongside corresponding DNI and DHI profiles generated from the TMY dataset using the BRL model (Lemos et al., 2017; Ridley et al., 2010). Despite differences in weather-type distributions caused by climate variability and extreme events, the GHI, DNI, and DHI curves show substantial agreement across both datasets. The total GHI errors for January and July were –6.0% and 1.7%, respectively; for DNI, –17.9% and –29.3%; and for DHI, 3.6% and 5.4%. While GHI and DHI showed relatively minor deviations, DNI exhibited a consistent negative bias in TMY-MER, especially during winter. Figure 9c compares the annual sky radiance models based on TMY-MER and TMY. The GHI sky models demonstrate strong spatial agreement across azimuthal directions, with TMY-MER slightly overestimating irradiance values, typically within 5%. The DNI distributions are also spatially aligned; however, TMY-MER systematically overestimates values, especially near the zenith, indicating greater summer overestimation. Additionally, the TMY-MER model shows higher DHI values in the northeastern sky sector compared to TMY, with the overestimation limited to under 20%.

# Calibrated solar radiation data separated into direct and diffuse radiation

**Figure 9.** Separation of direct and diffuse solar radiation components and error analysis; (a) Direct and diffuse separation of solar radiation data at P1814 station using TMY-MER; (b) Comparison of data following direct and diffuse separation for TMY-MER and TMY; (c) Sky radiation models following direct and diffuse separation for TMY-MER and TMY.

© Author(s) 2025. CC BY 4.0 License.

#### 4 Discussion

#### 4.1 Research contributions

This study has developed a high-resolution typical meteorological year dataset for the entirety of Australia, based on MERRA-2 reanalysis data, to address the spatial and temporal limitations of conventional TMY datasets in capturing localized meteorological variability (Gelaro et al., 2017). Unlike previous studies that merely identified systematic biases in MERRA-2-derived solar radiation (Cao et al., 2022; Du et al., 2022), this work systematically characterizes the spatiotemporal distribution and potential contributing factors of these errors, and implements targeted error assessment through an innovative methodology. By integrating data from 2009 to 2023, TMY-MER overcomes the reliance on outdated ground-based observations used in earlier datasets (Marion & Urban, 1995; Song et al., 2007). With enhanced spatial and temporal resolution, TMY-MER accurately captures the effects of terrain and microclimates on solar irradiance, offering reliable data for dynamic energy planning and urban thermal environment modelling.

A novel weather-type decomposition technique based on clear-sky identification was proposed to classify and correct radiation errors under different sky conditions. The analysis reveals that TMY-MER slightly overestimates solar radiation under clear-sky conditions, with annual mean errors below 5%. In contrast, under overcast conditions, errors become substantially larger, up to 40%, due to inaccuracies in cloud and aerosol modelling. These results support the hypothesis that cloud cover and aerosols are primary contributors to radiation bias, as suggested in previous studies (Cao et al., 2022; Du et al., 2022). Spatial analysis reveals that the most pronounced overestimations occur in densely populated southeastern coastal regions of Australia, with annual cumulative errors as high as 30%, while inland areas generally show smaller deviations, typically below 10%. From a seasonal perspective, the largest overestimations are observed during winter overcast periods, with errors reaching a maximum of around 30%, whereas errors under summer clear-sky conditions remain minimal and typically fall below 5%.

The integration of weather-type decomposition with IDW for parameter generalization enables TMY-MER to constrain total annual errors to within 15%. This error assessment framework ensures comprehensive spatial coverage across all 353 local government areas, addressing the challenge posed by the sparse distribution of conventional meteorological stations in Australia. Furthermore, the proposed methodology is both scalable and transferable, making it suitable for regions with complex climatic conditions and enhancing the global applicability of reanalysis datasets in both scientific research and practical planning contexts.

# 480 4.2 Discussion on error sources

This study systematically investigates the mechanisms contributing to errors in MERRA-2 solar radiation data through a spatiotemporal analysis. These biases primarily stem from the interplay between intrinsic limitations of the reanalysis model and the diverse environmental conditions across different regions of Australia. In industrialized eastern cities such as Sydney and Melbourne, elevated aerosol concentrations resulting from intensive anthropogenic activities suppress cloud droplet growth via the Twomey effect (Quaas et al., 2020), thereby prolonging cloud lifetime and enhancing cloud albedo (Jia et al., 2021). The absence of coupled aerosol-cloud dynamic feedback in MERRA-2 leads to systematic underestimation of cloud cover under overcast conditions, which in turn causes significant overestimation of solar radiation, reaching up to 40% in certain cases. This issue is particularly pronounced in urban areas during summer. For example, in Mount Gambier, SA, the lack of cumulus parameterization during the summer season results in radiation overestimations ranging from 20% to 40%, consistent with previous findings highlighting the dominant role of cloud and aerosol interactions (Du et al., 2022). In tropical monsoon regions such as northern QLD, wet-season (December to March) overcast conditions are dominated by deep convective systems. MERRA-2's simplified treatment of vertical cloud structure (Stamatis et al., 2022) likely

contributes to persistent underestimation of solar radiation during this period. In contrast, during the dry season, the model tends to underestimate low-level cloud cover, leading to excessive surface irradiance estimates (Camberlin et al., 2023).

**Figure 10.** Application of adjusted meteorological data in building cluster simulations: Comparison of solar radiation results from TMY and TMY-MER; (b) Surface solar radiation distribution for Building Cluster 2 based on TMY and TMY-MER; (c) Comparison of sky radiation model distribution characteristics.

Under clear-sky conditions, southeastern coastal regions generally show a modest positive bias of around 5%.

Meanwhile, overcast conditions in southwestern and eastern coastal areas are associated with more pronounced overestimations, with deviations reaching up to 20%. This discrepancy is likely due to the increased difficulty in accurately simulating radiative transfer processes under cloudy conditions. Urbanization further exacerbates these inaccuracies through the intensification of the urban heat island effect (Gao et al., 2025), which enhances local thermal gradients and convective activity. The resulting vertical moisture transport can trigger localized cumulus cloud formation when ascending air masses reach the condensation level. In densely populated areas, such clouds are often small in scale and are not adequately

represented by reanalysis systems due to simplifications in their physical parameterizations. As a result, these clouds frequently go undetected, introducing additional overestimations in radiative flux. Moreover, in temperate regions such as the southern highlands of VIC, solar radiation is frequently overestimated by 10% to 20%, possibly due to localized cloud

formation driven by diurnal temperature variations (Han et al., 2025), which is similarly not well captured by the MERRA-2 data

# 4.3 Data validation and application potential

To evaluate the applicability of the adjusted TMY-MER dataset, this study selected two representative building clusters with complex shading characteristics. Using both the unadjusted TMY and the adjusted TMY-MER datasets, monthly solar irradiance distributions were simulated at a spatial resolution of 0.5 m × 0.5 m. Cluster 1 (Fig. 10a) is located in Collie, WA. Irradiance on building surfaces was modelled using both datasets, and the results show strong agreement in both spatial distribution and intensity. The annual total deviation remained below 10%. For the representative months of each season March, June, September, and December, errors were generally under 10%, except in September, where the deviation increased to 16%. Cluster 2 (Fig. 10b), located in Brisbane, QLD, features a more complex urban geometry. Despite this, the two datasets produced closely matching irradiance patterns. The annual cumulative deviation did not exceed 5%, with winter deviations remaining below 5% and those in other months generally below 3%.

Figure 11. Database construction of multi-condition sky models; (a) Monthly radiation sky matrix; (b) Typical day radiation sky matrix.

Figure 10c illustrates the sky radiation models based on TMY-MER and TMY datasets for Cluster 1 and Cluster 2.

Notable differences in sky radiance distribution can be observed between the two datasets. For Cluster 1, discrepancies during winter are primarily reflected in the irradiance magnitude, whereas variations in the transitional and summer seasons are more apparent in spatial distribution. In contrast, the sky radiance models for Cluster 2 show nearly identical spatial layouts across all seasons, with only minor differences in intensity. After error assessment, TMY-MER tends to slightly overestimate surface solar irradiance compared to TMY. Despite this, the TMY-MER-simulated spatial distribution of surface irradiance closely matches that of TMY, with differences remaining within acceptable uncertainty margins. These deviations are likely due to climate variability. The observed consistency in luminance patterns and magnitudes in the sky

models derived from both datasets provides compelling validation of the accuracy and reliability of the adjusted TMY-MER dataset.

TMY-MER can be employed as a high-resolution, globally available TMY dataset. It serves as a viable alternative to conventional TMY data for solar radiation modelling applications. With improved spatial resolution and broader geographic coverage, it provides a more accurate and efficient data foundation for scientific research and engineering applications. This, in turn, promotes technological innovation and advancements in solar energy utilization and related fields, underscoring its significant theoretical and practical value. Figure 11a illustrates the development of a comprehensive set of 36 numerical sky radiance models based on monthly GHI, DNI, and DHI data, addressing the growing demand for assessments across various temporal scales, such as monthly evaluations and characteristic days. Figure 11b shows another set of 36 models constructed to represent conditions around the solar equinoxes and solstices, including the days immediately preceding and following each event. This multi-dimensional modelling framework effectively addresses limitations in previous studies, which often lacked the capacity to simulate complex radiative scenarios on a case-by-case basis. By selecting and combining appropriate sky radiance models based on specific computational requirements, researchers can efficiently evaluate solar energy distributions under diverse conditions within a single simulation run. For instance, this includes monthly analyses of total, direct, and diffuse solar radiation on building surfaces in energy studies, or assessments of total solar exposure during representative winter and summer days in urban thermal environment studies.

# 5 Data availability

The TMY-MER database can be accessed in the Zenodo dataset from the following DOI: https://doi.org/10.5281/zenodo.15479502 (Fu et al., 2025).

# 6 Conclusions

This study developed a high-resolution TMY dataset, termed TMY-MER, based on MERRA-2 reanalysis data and covering the entire Australian continent. To address systematic errors inherent in reanalysis datasets and those arising from climate variability, a calibration approach was implemented through clear-sky identification and weather type classification, further refined using spatial interpolation techniques to enhance data accuracy. Validation results demonstrate that the calibrated dataset exhibits high reliability in complex urban environment simulations, offering substantial support for dynamic energy planning and urban climate research.

- (1) Under clear-sky conditions, the TMY-MER dataset demonstrated a slight positive bias in solar radiation estimates, with annual mean errors typically below 5%. In contrast, overcast conditions introduced substantially larger deviations, primarily due to inaccuracies in cloud and aerosol representation, with maximum errors reaching 40%. Spatially, the most significant overestimations were observed in densely urbanized southeastern coastal regions, where typical annual biases ranged from 10% to 15%, and localized peaks approached 30%. Inland areas showed smaller errors, mostly between 5% and 10%. From a temporal perspective, the largest deviations occurred during winter overcast periods, while summer clear-sky conditions exhibited consistently high accuracy.
- (2) Decomposition of GHI into DNI and DHI using the DIRINT model revealed a systematic underestimation in DNI, with monthly biases of -17.9% in January and -29.3% in July, and a consistent overestimation in DHI ranging from 3.6% to 5.4%. Hourly irradiance patterns and sky dome distributions remained highly consistent across datasets, confirming the strong temporal fidelity of TMY-MER in capturing diurnal solar variation.
  - (3) Based on the adjusted TMY-MER data, a suite of 36 numerical sky radiance models was constructed to represent monthly and typical-day conditions, incorporating GHI, DNI, and DHI components. This model library enables efficient and

high-resolution solar simulations under diverse environmental scenarios, offering robust analytical support for building energy modelling, urban thermal environment evaluation, and photovoltaic system optimization.

#### Author contributions.

JF and PN contributed equally to this work. JF and PN conceived the research idea, developed the methodology, collected and processed the MERRA-2 reanalysis data, and drafted the manuscript with revisions. ZY and BH supervised the research, secured funding, and managed project resources. DP validated data through building simulations and provided critical feedback. GQ and FL implemented spatial interpolation and software support. YY and JY performed weather-type classification and data quality checks. All authors participated in writing, reviewing, and editing the manuscript.

# Competing interests.

The authors declare that they have no conflict of interest.

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
