# Peer review of "Developing a High-Resolution Typical Meteorological Year Dataset for Solar Radiation Evaluation in Australia"

_Earth System Science Data, 2025_

## Referee Comment (RC1)

**Review of "Developing a High-Resolution Typical Meteorological Year Dataset for Solar Radiation Evaluation"**

**Summary-** In this paper, the authors present a new typical meteorological year dataset (TMY-MER) related to solar radiation in Australia. The innovation here is that the authors have used reanalysis data on solar radiation from MERRA-2. This data was transformed into a TMY dataset. Since solar radiation data is sensitive to sky conditions and weather, a weather classification-based adjustment is applied to reduce errors. Finally, this is combined with an IDW interpolation method to generate transpose coarser errors from 122 sites to 353 sites. The authors present detailed error comparisons between this new "TMY-MER-IDW" dataset and original TMY observational data. This paper deals with a relevant and important topic. But I had several questions for the authors regarding descriptions in the paper. I find their error metrics a bit confusing, and I believe there are several nuances here missing description. I therefore recommend major revisions. I have summarized my comments below.

**General comments-**

1) **This paper really deals with two distinct datasets (with and without the IDW)-** Reading through this paper I realized that the authors have in fact created two distinct datasets, the first is the TMY-MER dataset itself (which has no error adjustment) and then the TMY-MER-IDW dataset where they apply the IDW method to transpose errors from observational stations to 353 stations in question. This becomes especially clear in the results section (Pg 16 and Figure 7) where the errors are significantly reduced when looking at the IDW based results. However, the methodology descriptions lumps in the description of the IDW step with other steps which would not be accurate given that this single step seems to change errors so much. As a simple example, Step 5 on Page 8 should really be its own section. I recommend authors distinguish between the datasets with and without the IDW throughout the manuscript to ensure this point is conveyed correctly.

2) **Description of generating the TMY-MER-**I must confess that I was really lost when reading the section on the description of the TMY-MER dataset construction (Page 6 and 7). I think unless a user is intimately familiar with how TMY data is constructed, they will be lost as well. Here are my main points related to this-
   a. Firstly, the authors make use of 7 key parameters and monthly values are extracted for each. Why would these parametric data be needed when the GHI data are already available in the MER data?
   b. On line 179, why is this criteria necessary (all normalized values being less than 1)?
   c. On line 181, once again, why is a weighting even necessary when GHI data is available as a separate variable at all?

     **d.** Just a note that the authors are suddenly referring to an "AMY" dataset that that has not been defined before in point number 1. Is this just the MERRA-2 data?

     **e.** Authors mention 7 variables, but there are 8 (excluding the GHI) in Table 1?

**3) Discussion of error metrics-** The error metrics discussed in this paper (e.g. Figure 2, Figure 5) concern the radiation from the TMY-MER vs a regular TMY dataset. Just want to confirm that this is observational data from sites? I'm guessing it is. If so, I think what would really be required here would be comparing TMY-MER data to similar data from another source in addition to the observational data? For example, of all of the datasets discussed on lines 74-81, the authors could select a couple and describe how their dataset is better than what's available in the literature. If authors are comparing against observational data only, I'm not sure what to make of the "errors", since I don't know whether there are better performing reanalysis data already available. In addition to this, it might be worth showing errors in a step-by-step manner i.e. first just the raw TMY-MER, then the TMY-MER with weather classification and finally the TMY-MER-IDW dataset (Authors already discuss the third one separately). This would give an idea of the step-by-step improvement.

**4) Descriptions of existing data-** As discussed above, the authors summarize existing data on lines 76-81 on Page 3. However, this description does not make clear what these alternative methods are or how what the authors have constructed is different. I would recommend they make a table where they list each method and then summarize what these methods are. That will make the differentiation of the author's contribution far easier to understand. Again, once again I believe that this needs to be written for someone not familiar with TMY data i.e. the general reader.

**5) Alternative comparisons for Figure 3 and 4-** There seem to be some significant differences in the distributions here between the TMY and TMY-MER. Would it be possible to show this as a scatterplot to illustrate the similarities. You can keep the monthly facet as is and just color the dots? Would it also be possible to show some summary statistics for these distributions, maybe a SD value across all distributions in a facet?

**Other points-**

1) As mentioned in the points above, in Figure 1, it would be helpful to separate out the IDW step as a separate step rather than combined in step 2. This particular step by itself reduced errors greatly.

2) Line 174- As mentioned above, authors refer to an "AMY" dataset that is never defined. This is just the MERRA2 data, right?

3) Line 211- What is the threshold value? How was it selected and was there a sensitivity conducted for this parameter?

4) Lines 255-259- While the application of this dataset using the DIRINT model is appreciated, I believe this adds a layer of uncertainty since this model seems to be driven by its own parameters and their uncertainties. The results from DIRINT are also briefly discussed in a small section (3.6). Can this all be moved to the SI with a simple summary in the main text? The real validation in this manuscript is the error analysis.